# Impact of body mass index on mortality outcomes in intensive care patients with *Staphylococcus aureus* sepsis: A retrospective analysis

Heping Xu[1]*, Yiqiao Liu[2], Huan Niu[1], Hong Wang[1], Feng Zhan[1]

1 Department of Emergency Medicine, Hainan General Hospital/ Hainan Affiliated Hospital of Hainan Medical University, Haikou, Hainan, China, 2 Department of Emergency Medicine, Hainan Affiliated Hospital of Hainan Medical University, Haikou, Hainan, China

☯ These authors contributed equally to this work.
* xhp21528@163.com

## Abstract

### Background

Evidence associating body mass index (BMI) with the prognosis of *Staphylococcus aureus* sepsis remains scarce.

### Objective

To explore the association between BMI and clinical outcomes in intensive care units patients with *Staphylococcus aureus* sepsis.

### Methods

A retrospective analysis of patients with *Staphylococcus aureus* sepsis was conducted using the MIMIC-IV database from the Critical Care Medicine Information. Data were collected within the first 24 hours of intensive care units admission. The primary endpoint was 28-day mortality. The association between BMI and 28-day all-cause mortality was assessed using multivariable logistic regression, subgroup analyses, restricted cubic spline curves and Kaplan-Meier survival analysis.

### Results

The study included 2,295 patients with an average age of 63.5 (16.1) years, 60.2% of whom were male. Multivariate analysis revealed that each 1 kg/m$^2$ increase in BMI was linked to a 2.8% decrease in the risk of 28-day mortality (adjusted OR = 0.972, 95% CI: 0.955–0.990, P = 0.002). Patients in the medium and high BMI categories had significantly lower risks of 28-day mortality compared to those in the low BMI group (OR [95% CI] 0.650 [0.474–0.891]; OR [95% CI] 0.516 [0.378–0.705]; P trend < 0.0001). The RCS model showed a non-linear association between BMI and 28-day mortality (P = 0.014). Kaplan-Meier analysis showed

**Data Availability Statement:** The datasets generated and/or analyzed during the current study are available in the MIMIC-IV database (https://mimic.physionet.org/). Access to the MIMIC-IV

database is available through the PhysioNet platform (https://physionet.org/content/mimiciv), subject to registration and completion of the required training in human subjects research (CITI program). Due to the sensitive nature of the data, direct sharing of the dataset is not permissible. Interested researchers can obtain the data directly from PhysioNet following the necessary approval process.

**Funding:** This work was supported by Hainan Provincial Natural Science Foundation of China. Project 823RC560. The funders had no role in study design, data collection and analysis, decision to publish, or preparation of the manuscript.

**Competing interests:** The authors declare that they have no competing interests.

that patients with elevated BMI had lower 28-day mortality (P < 0.0001). Notably, significant interactions between AKI and SOFA with BMI were observed (P<0.05).

## Conclusion

Increased BMI is associated with a reduced risk of 28-day all-cause mortality in patients with *Staphylococcus aureus* sepsis.

## Introduction

In recent years, the global incidence and prevalence of sepsis have increased significantly [1,2]. Concurrently, obesity remains a prevalent issue worldwide, particularly in the United States where over 25% of adults admitted to intensive care units (ICUs) are classified as overweight, obese, or morbidly obese based on their body mass index (BMI) [3–6]. Sepsis is a common reason for these admissions [7–9].

Despite the high prevalence of both conditions, studies exploring the relationship between BMI and the prognosis of sepsis patients are limited. This gap in research highlights the need to investigate the link between BMI and 28-day mortality in patients with sepsis. While obesity is typically associated with increased cardiovascular risks, it may paradoxically offer protective benefits against certain conditions, a phenomenon referred to as the 'obesity paradox.' Observational studies suggest that a higher BMI upon admission is correlated with improved survival and functional outcomes in sepsis patients [10]. *Staphylococcus aureus* sepsis, representing the most severe form of *Staphylococcus aureus* infection, accounts for nearly 20% of all hospital-acquired sepsis cases [11,12]. However, the evidence supporting the obesity paradox in cases of *Staphylococcus aureus* sepsis is still emerging.

Therefore, this study aims to explore the relationship between BMI and 28-day mortality in patients suffering from *Staphylococcus aureus* sepsis, which is critical for enhancing clinical decision-making and understanding patient prognosis.

## Materials and methods

### Database introduction

The study cohort consisted of ICU-admitted patients registered in the MIMIC-IV database (V2.2) (DOI: 10.13026/6mm1-ek67) [13]. This registry was developed from sophisticated multi-parameter monitoring systems at the Beth Israel Deaconess Medical Center (BIDMC) in Boston, Massachusetts, encompassing detailed records of over 50,000 patients admitted from 2008 to 2019 [14].

The MIMIC-IV database encompasses extensive data from various Intensive Care Units (ICUs) such as the Medical ICU, Cardiac Surgery ICU, Surgical ICU, Cardiology ICU, Trauma Surgery ICU, and Neonatal ICU. This data includes demographic information, vital signs, laboratory test results, and diagnosis codes consistent with the ICD-9 and ICD-10 classifications. Author Xu, having completed the Citi Program online training course (record ID 59568270), extracted data using the PostgreSQL tool.

MIMIC-IV is an anonymized public database approved by the institutional review boards of the Massachusetts Institute of Technology (MIT) and Beth Israel Deaconess Medical Center (BIDMC). The requirement for informed consent was waived due to the thorough anonymization and de-identification of all patient information in the database.

## Definitions

BMI [15] is calculated by dividing a person's weight in kilograms by their height in meters squared. Three categories were defined: BMI1 (BMI <25 kg/m$^2$); BMI2 (BMI 25–30 kg/m$^2$); and BMI3 (BMI ≥30 kg/m$^2$). The primary focus was on 28-day mortality, with secondary parameters including in-hospital mortality, length of stay (LOS), and ICU duration. Acute Kidney Injury (AKI) is defined as: an increase in serum creatinine of ≥ 0.3 mg/dL (≥ 26.5 μmol/L) within 48 hours, or an increase in serum creatinine to ≥ 1.5 times the baseline level within the past week, or a reduction in urine output to < 0.5 mL/kg/h for 6 hours [16]. Shock is defined by the need for vasoactive drugs during an ICU stay. According to sepsis-3 criteria, organ dysfunction was assessed using the SOFA score, with ≥2 indicating dysfunction [17]. Methicillin-resistant Staphylococcus aureus (MRSA) infection refers to any positive culture for MRSA, whether it is from blood, wound, sputum, or any other body site.

## Study population and data collection

The inclusion criteria for this study encompassed patients aged 18 years and older who were diagnosed with *Staphylococcus aureus* sepsis, during their ICU stay. This research incorporated a comprehensive cohort diagnosed with *Staphylococcus aureus*, using both ICD-9 and ICD-10 codes. Specifically, diagnoses were identified using codes such as '03811', '03812', '04111', '04112', '48241', '48242', 'A4101', 'A4102', 'A4901', 'A4902', 'Z22321', 'Z22322', 'B9561', 'B9562', 'V0253', 'V0254', 'J15211', 'J15212'. Only data from each patient's initial ICU admission were included. Additionally, all patients were diagnosed using the Sepsis-3 criteria. Exclusion criteria were: (1) patients not on their first hospital or first ICU admission to prevent duplications; (2) patients with a hospital or survival time less than 24 hours to ensure adequate time for assessing their conditions and outcomes; and (3) those lacking height or weight data, essential for accurate BMI calculation.

The primary outcome measured was 28-day mortality. Extracted data included variables such as age, gender, height, weight, comorbidities, Charlson Comorbidity Index (CCI), day one Sequential Organ Failure Assessment (SOFA) score, Simplified Acute Physiology Score II (SAPS II), and initial ICU admission laboratory results (complete blood count, coagulation profile, glucose, urea nitrogen, creatinine, calcium, lactate). Other crucial data points considered included continuous renal replacement therapy, invasive ventilation, the presence of septic shock, hospital length of stay, ICU length of stay, and hospital mortality.

The data for this study were derived directly from critical care information systems, electronic hospital records, laboratory results, and vital sign monitors. The required data can be downloaded and analyzed. Notably, the MIMIC-IV database, used in this study, is linked with the US Social Security System to facilitate tracking of post-discharge mortality data [14].

## Statistical analysis

Data analysis was conducted using Stata version 18.0. Continuous variables were reported as mean (standard deviation) or median (interquartile range), and categorical variables were expressed as percentages. The baseline characteristics across various BMI categories were assessed using the chi-square test for categorical data, one-way analysis of variance for normally distributed continuous data, and the Kruskal-Wallis H test for non-normally distributed data. The association between BMI and 28-day all-cause mortality was examined through multivariate logistic regression. The variation inflation factor (VIF) values were used to evaluate multicollinearity, with VIFs exceeding 10 indicating significant multicollinearity. We developed three models: an unadjusted model 1, a model 2 adjusted for age and sex, and a model 3 adjusted for all variables. Using stepwise forward regression, we included variables with a

significance threshold of P < 0.05 and excluded those with a significance threshold of P > 0.1. The final model 3 included the following variables: age, diabetes without control, cerebrovascular disease, the first day SOFA score, white blood cell (WBC), hemoglobin, platelets, red cell distribution width (RDW), creatinine, glucose, blood urea nitrogen (BUN), lactate, invasive ventilation, septic shock.

For the sensitivity analysis, interactions and subgroup analyses were conducted based on factors such as age (<75 and ≥75 years), sex, MRSA status, congestive heart failure, chronic pulmonary disease, uncontrolled diabetes, cerebrovascular disease, renal disease, AKI, CRRT, invasive ventilation, SAPS II scores (<37 and ≥37), SOFA scores (<4 and ≥4), and CCI presence.

Restricted cubic splines (RCS) were used to determine critical values and visualize the non-linear association between BMI at ICU admission and 28-day mortality. Kaplan-Meier survival analyses were used to compare the 28-day survival rates of ICU patients with varying BMI levels. Statistical significance was determined at P< 0.05. Data were analyzed using the R statistical software package (version 4.1.1)

## Results

### Baseline characteristics of the included participants

A total of 2,295 patients from the MIMIC-IV database met the inclusion criteria, as illustrated in Fig 1. The baseline characteristics of the patients across the BMI categories are detailed in Table 1. The average age of these patients was 63.5 years with a standard deviation of 16.1, and approximately 60.2% were male. The patients were divided into three groups based on their body mass index: BMI1, BMI2, and BMI3. No significant differences were observed among the groups in terms of MRSA, renal disease, Charlson Comorbidity Index, SOFA score, SAPS II, WBC count, hemoglobin, platelet levels, BUN, calcium, lactate, use of invasive ventilation, septic shock, ICU length of stay, or hospital length of stay (all P > 0.05). Compared to patients with the lowest BMI, those with the highest BMI are more likely to have comorbidities such as congestive heart failure, chronic lung disease, diabetes without controlled, renal disease, and acute kidney injury. They are also more likely to require continuous renal replacement therapy (CRRT). As BMI increases, both in-hospital mortality and 28-day mortality decrease, as do the average age and proportion of male patients.

### Results of logistic regression

Three distinct logistic regression models were developed to evaluate the independent effect of BMI on 28-day all-cause mortality among ICU patients with *Staphylococcus aureus* sepsis (Table 2). Logistic regression analysis indicated an inverse association between BMI and the risk of 28-day all-cause mortality, with an unadjusted odds ratio (OR) of 0.968 (95% CI 0.953–0.983). Using BMI1 as the reference, the crude ORs for BMI2 and BMI3 groups were 0.723 (95% CI 0.551–0.948) and 0.531 (95% CI 0.407–0.692), respectively. After adjusting for age and sex, this inverse association persisted (OR 0.975; 95% CI 0.960–0.991), with adjusted ORs for BMI2 and BMI3 at 0.699 (95% CI 0.529–0.923) and 0.589 (95% CI 0.499–0.773), respectively. Model 3, which incorporated adjustments for age, diabetes without controlled, cerebrovascular disease, first day SOFA score, WBC count, hemoglobin level, platelet count, RDW, creatinine level, glucose level, BUN, lactate level, invasive ventilation, septic shock, showed BMI maintained a strong inverse correlation with 28-day all-cause mortality (OR 0.972; 95% CI 0.955–0.990). The ORs for BMI2 and BMI3 in this model were 0.650 (95% CI 0.474–0.891) and 0.516 (95% CI 0.378–0.705), respectively. Linear trend tests across the models underscored a consistent association between BMI and 28-day mortality.

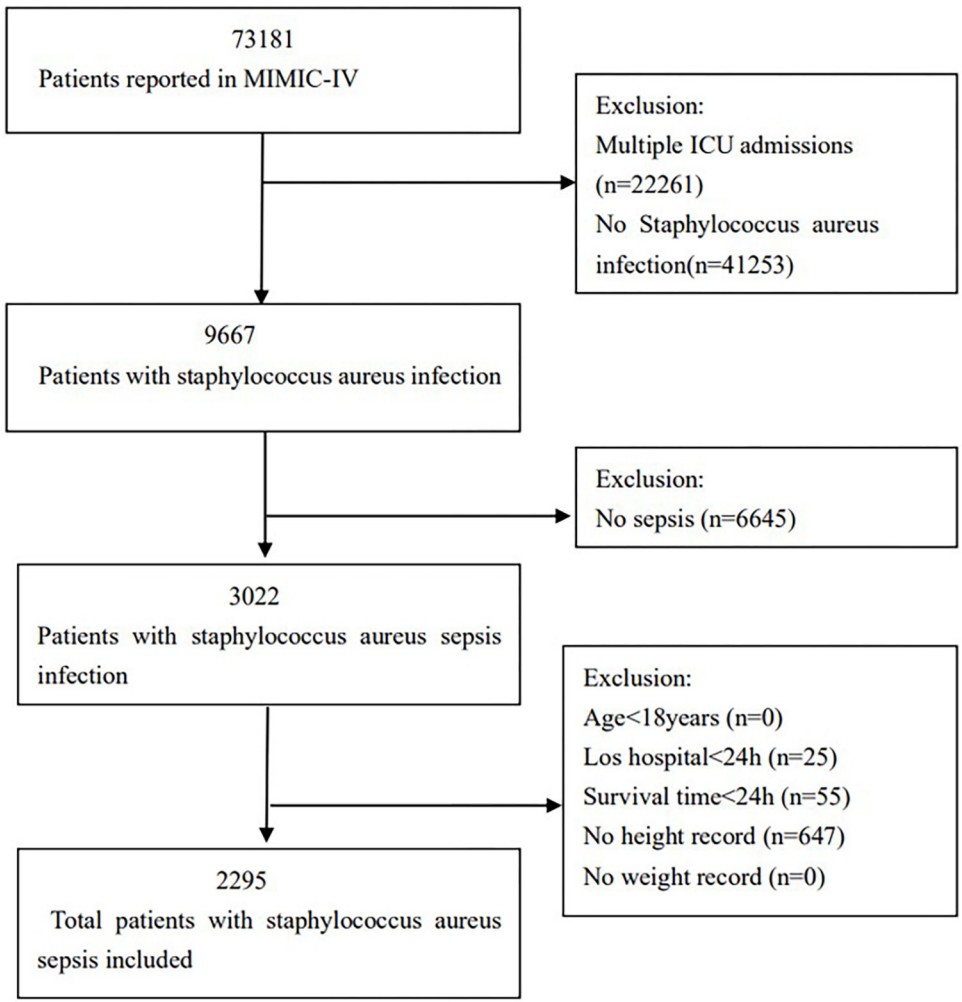

**Fig 1. Flow chart illustrating the selection of patients included in the analysis.**

## Results of RCS and Kaplan-Meier analysis

Using restricted cubic splines to determine thresholds, we visualized the nonlinear association between BMI at ICU admission and 28-day mortality, as shown in Fig 2. The analysis revealed a significant nonlinear association between BMI at ICU admission and 28-day mortality in sepsis patients (P = 0.014). When BMI exceeded 28.24 kg/m$^2$, the odds ratio for mortality rapidly decreased, but with further increases in BMI, the odds ratio leveled off. Overall, the graph exhibits an "L" shape, indicating that higher BMI levels at ICU admission are associated with lower 28-day mortality odds ratios. Furthermore, Kaplan-Meier curves for different BMI subgroups are presented in Fig 3. The results indicate that higher BMI is associated with lower 28-day mortality (log-rank test, P < 0.0001).

## Results of subgroup analyses

To explore potential clinical heterogeneity, we employed interaction and stratification analyses (Fig 4). We examined the association between BMI and 28-day mortality across various subgroups. No significant interactions or differences were observed in stratified analyses based on age (<75 and ≥75 years), sex, MRSA status, renal disease, congestive heart failure, diabetes

**Table 1. Demographic and clinical characteristics of patients with *staphylococcus aureus* sepsis according to body mass index.**

| Characteristics | All patients (N = 2295) | BMI | | | P |
| --- | --- | --- | --- | --- | --- |
| | | BMI <25 (N = 697) | 25≤BMI<30 (N = 677) | BMI≥30 (N = 921) | |
| Age, years (mean standard deviation) | 63.5 ±16.1 | 64.1±18.1 | 65.1±16.3 | 61.9 ±14.0 | <0.0001 |
| Men | 1381(60.2) | 410(58.8) | 449(66.3) | 522(56.7) | <0.0001 |
| MRSA | 986(43.0) | 311(44.6) | 285(42.1) | 390(42.3) | 0.568 |
| Comorbidities | | | | | |
| Congestive heart failure | 713(31.1) | 184(26.4) | 214(31.6) | 315(34.2) | 0.003 |
| Chronic pulmonary disease | 614(26.8) | 172(24.7) | 156(23.0)- | 286(31.1) | 0.001 |
| Diabetes without control | 658(28.7) | 140(20.1) | 185(27.3) | 333(36.1) | <0.0001 |
| Cerebrovascular disease | 335(14.6) | 113 (16.2) | 108(16.0) | 114(12.4) | 0.047 |
| Renal disease | 650(28.3) | 194(27.8) | 199(29.4) | 257(27.9) | 0.762 |
| Score | | | | | |
| Charlson comorbidity index median (IQR)) | 6(4–8) | 6(4–7) | 6(4–8) | 6(4–7) | 0.134 |
| First day of SOFA, median (IQR) | 6(4–9) | 6(3–9) | 6(4–9) | 6(4–9) | 0.291 |
| SAPSII | 37(29–48) | 37(29–48) | 38(30–48) | 37(29–47) | 0.322 |
| Laboratory variables at admission, mean (IQR) | | | | | |
| WBC, cell/mm3 | 9.0(6.8–12.3) | 9.0(6.6–12.3) | 9.0(6.9–12.3) | 9.1(6.9–12.3) | 0.186 |
| Hemoglobin, mg/dL | 11.2(9.4–12.9) | 11.2(9.5–12.8) | 11.2(9.5–13.0) | 11.2(9.4–12.8) | 0.776 |
| Platelet, cell/mm3 | 235(165–309) | 235(159–308) | 235(165–306) | 235(169–312) | 0.701 |
| RDW, % | 14.7(13.7–16.2) | 14.7(13.6–16.0) | 14.7(13.8–16.0) | 14.7(13.8–16.5) | 0.014 |
| PT, sec | 13.3(12.2–15.2) | 13.1(12.0–14.6) | 13.3(12.3–15.6) | 13.3(12.2–15.6) | 0.0004 |
| Creatinine, mg/dL | 1.0(0.8–1.5) | 1.0(0.7–1.4) | 1.0(0.8–1.5) | 1.0(0.8–1.5) | 0.004 |
| Glucose, mg/dL | 122(100–157) | 115(95–148) | 122(99–153) | 127(104–164) | 0.0001 |
| BUN, mg/dL | 21(14–34) | 21(13–32) | 21(15–35) | 22(15–35) | 0.100 |
| Calcium, m Eq/L | 8.6(8.1–9.2) | 8.7(8.1–9.2) | 8.6(8.1–9.1) | 8.6(8.1–9.2) | 0.954 |
| Lactate, mmol | 1.5(1.1–2.1) | 1.5(1.1–2.1) | 1.5(1.1–2.0) | 1.5(1.1–2.1) | 0.400 |
| Interventions | | | | | |
| Invasive ventilation | 557(24.3) | 187(26.8) | 165(24.3) | 205(22.2) | 0.105 |
| CRRT | 59(2.6) | 8(1.1) | 17(2.5) | 34(3.7) | 0.006 |
| Outcomes | | | | | |
| Septic shock | 872(38.0) | 273(39.2) | 245(36.2) | 354(37.5) | 0.492 |
| AKI | 1537(67.0) | 398(57.1) | 428(63.2) | 711(77.2) | <0.0001 |
| LOS ICU, median (IQR)) | 3.9 (1.9–9.2) | 3.9(1.8–9.1) | 3.9(1.9–8.6) | 4.0 (1.9–9.9) | 0.266 |
| LOS hospital, median (IQR)) | 13.0(7.2–22.1) | 12.6(7.0–22.4) | 12.5(7.1–20.6) | 13.6(7.6–22.5) | 0.096 |
| In-hospital mortality | 360(15.9) | 138(19.8) | 99(14.6) | 123(13.4) | 0.001 |
| 28-day mortality | 379(16.5) | 150(21.5) | 112(16.5)) | 117(12.7) | <0.0001 |

BMI, Body mass index; WBC, White blood cells; RBC, Red blood cell; PT prothrombin times; RDW, Red cell distribution width; BUN, Blood urea nitrogen; CRRT, Continuous renal replacement therapy; AKI, Acute kidney injury; LOS, Length of stay. SOFA, Sequential organ failure assessment.

without control, chronic pulmonary disease, invasive ventilation septic shock, SAPS II score (<37 and ≥37), CCI (<7 and ≥7). Notably, significant interactions between AKI and SOFA with BMI were observed (P<0.05). Non-AKI patients with a higher BMI had a lower risk of death compared to those with AKI. Additionally, patients with a higher BMI and a SOFA score of less than 4 had a lower risk of death than those with a SOFA score of 4 or greater.

**Table 2. Relationship between BMI and 28-day all-cause mortality in different models.**

| | Model1 OR (95% CI) P value | | Model2 OR (95% CI) P value | | Model3 OR (95% CI) P value | |
|---|---|---|---|---|---|---|
| BMI | 0.968 (0.953,0.983) | <0.0001 | 0.975(0.960,0.991) | 0.002 | 0.972(0.955,0.990) | 0.002 |
| BMI | | | | | | |
| BMI1 | Ref | | Ref | | Ref | |
| BMI2 | 0.723(0.551,0.948) | 0.019 | 0.699(0.529,0.923) | 0.012 | 0.650(0.474,0.891) | 0.005 |
| BMI3 | 0.531(0.407,0.692) | <0.0001 | 0.589(0.499,0.773) | <0.0001 | 0.516(0.378,0.705) | <0.0001 |
| P for trend | <0.0001 | | <0.0001 | | <0.0001 | |

OR, odds ratio; CI, confidence interval; Ref, reference; BMI, body mass index MRSA Methicillin-resistant *Staphylococcus aureus*, Model 1 was not adjusted; Model 2 was adjusted for age, sex; and Model 3 was adjusted for age, Diabetes without control, Cerebrovascular disease, First day of SOFA, WBC, Hemoglobin, Platelet, RDW, Creatinine, Glucose, BUN, Lactate, Invasive ventilation, Septic shock.

## Discussion

In our study, we examined the association between BMI and clinical outcomes in patients with *Staphylococcus aureus* sepsis admitted to the intensive care unit. Our analysis identified a clear association between BMI and 28-day mortality rates. The data prominently demonstrated an L-shaped curve association between BMI and 28-day mortality, indicating a negative correlation and embodying the "obesity paradox" wherein obese patients demonstrated a higher

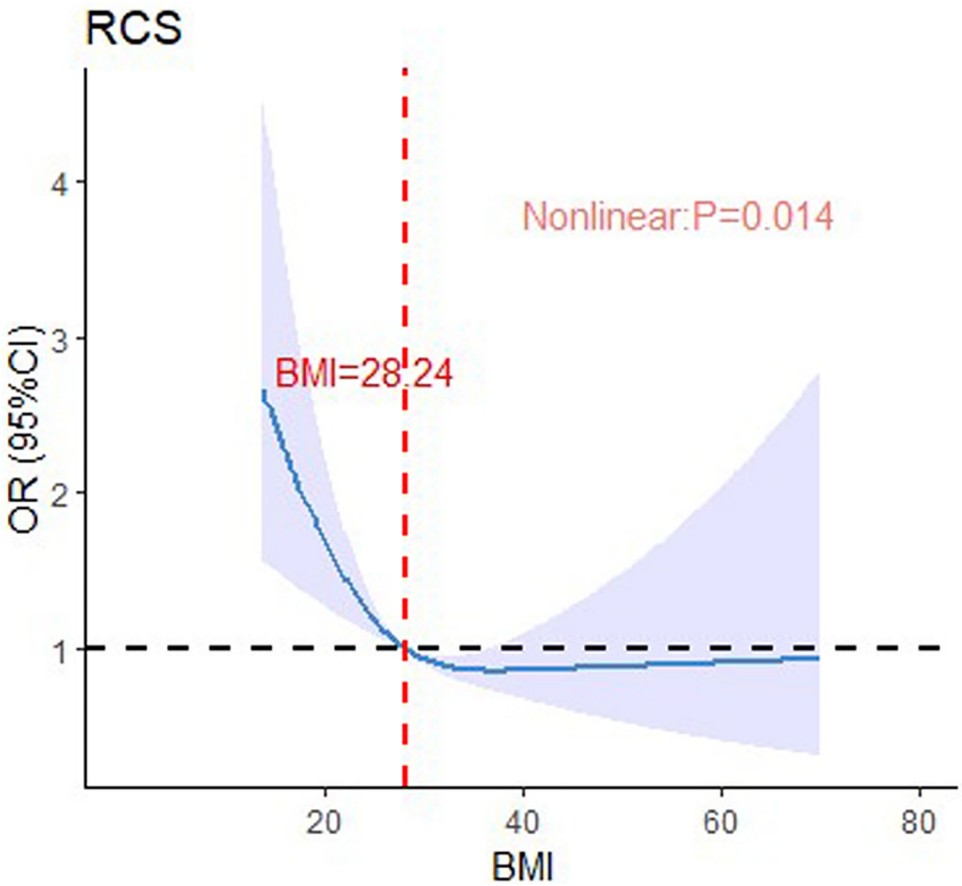

**Fig 2. Nonlinear relationship between the BMI and 28-day mortality.**

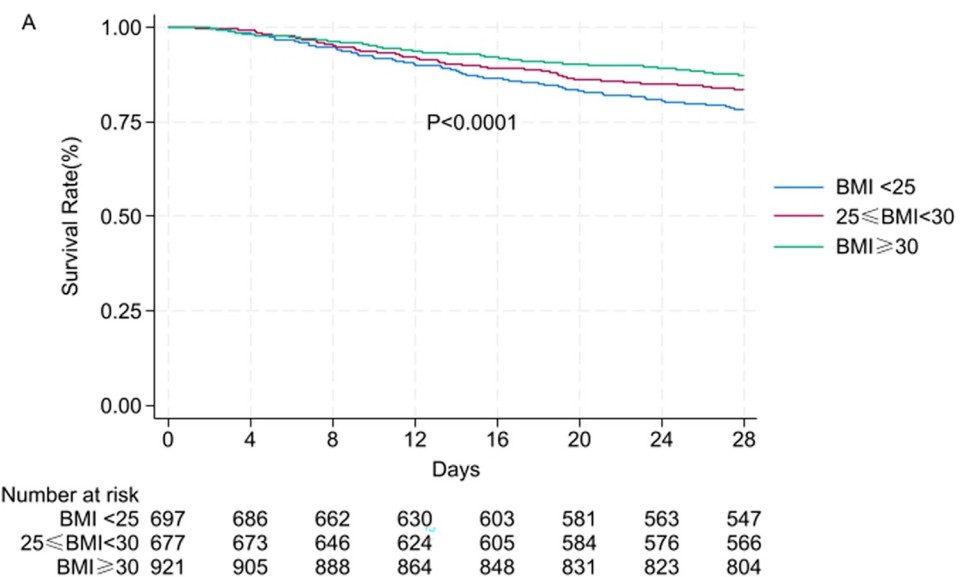

**Fig 3. 28-day survival rate of *Staphylococcus aureus* sepsis patients classified by BMI using the Kaplan-Meier method.**

survival rate, whereas underweight patients were at an increased risk of mortality. Notably, when BMI exceeded 28.24 kg/m$^2$, the odds ratio for mortality rapidly decreased, however, with further increases in BMI, the odds ratio leveled off.

The phenomenon of the obesity paradox is well-documented in both short-term and long-term survival outcomes for sepsis patients [18]. A plethora of studies have explored the influence of varying BMI subgroups on mortality rates in patients hospitalized with bacteremia, sepsis, and septic shock [19,20]. A large cohort study involving 55,038 adult patients across 139 U.S. hospitals highlighted that higher BMIs correlated with lower short-term mortality in both unadjusted and adjusted analyses [21]. Similarly, a meta-analysis indicated that overweight or obese ICU patients with sepsis or septic shock experienced reduced adjusted mortality rates [22], which corroborates our findings. However, specific clinical outcomes relating BMI to *Staphylococcus aureus* sepsis remain sparsely reported. In this study, we focused on the short-term mortality of patients with *Staphylococcus aureus* sepsis, specifically within 28 days. Our analysis showed that underweight patients had a significantly increased risk of death. This may be because underweight patients are more likely to experience metabolic imbalances and malnutrition, which can lead to a weakened immune system and overall poorer health, making it more difficult for them to effectively combat severe infections. Similar findings were observed in the study by Lin et al [23].

Previous studies have indicated that MRSA is a significant risk factor for short-term mortality in severe sepsis [24,25]. Therefore, in this study, we included it as a confounding factor for consideration. However, our findings revealed that the proportion of MRSA did not significantly differ between different BMI groups. This suggests that MRSA positivity is not a major confounding factor in the relationship between BMI and 28-day mortality. Further analysis showed that after controlling for other potential confounders, for each unit increase in BMI, the 28-day mortality decreased by 2.8%. This suggests that a higher BMI provides some protective effect for patients. This phenomenon partly explains the "obesity paradox," where a higher BMI is associated with better outcomes in certain severe illnesses. However, this protective effect is not absolute and can be influenced by other clinical factors.

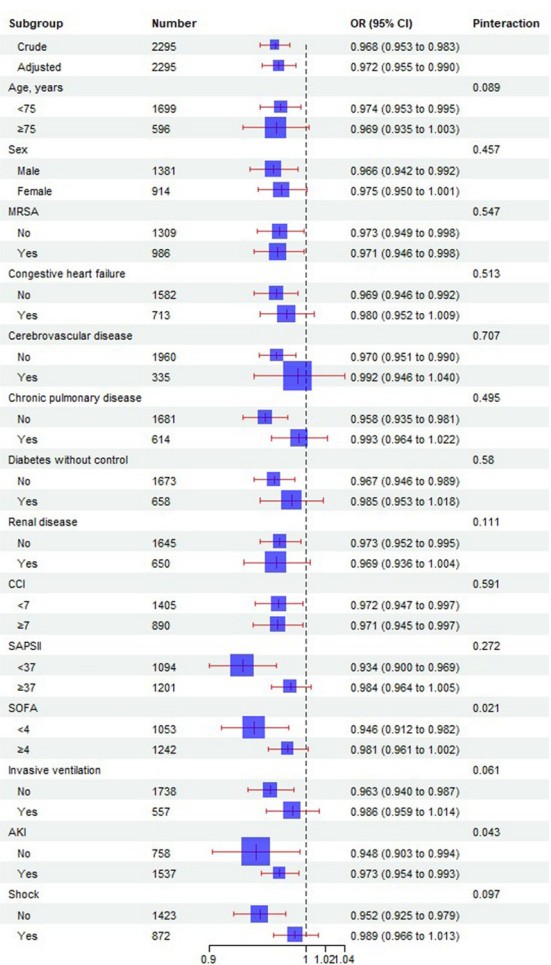

**Fig 4. Effect size of BMI on 28-day mortality in prespecified and exploratory subgroups.** The effect size was adjusted for age, Diabetes without control, Cerebrovascular disease, First day of SOFA, WBC, Hemoglobin, Platelet, RDW, Creatinine, Glucose, BUN, Lactate, Invasive ventilation, Septic shock.

For instance, AKI and the SOFA score play important roles in modulating the association between BMI and mortality risk. Our research found that patients with higher BMI and without AKI had a significantly lower risk of death compared to those with AKI. This may be because AKI itself is a serious complication often associated with poorer prognosis, and heavier patients might better mitigate the metabolic burdens associated with AKI.

Furthermore, the SOFA score significantly affects the association between BMI and mortality. For patients with a SOFA score below 4, a higher BMI is associated with a lower risk of death. In contrast, in cases of multiple organ failure (SOFA score ≥ 4), this protective effect is weakened or even disappears. This indicates that when organ function is not severely compromised, a higher BMI may offer protective benefits; however, in situations of multiple organ failure, this protective effect might no longer exist.

These findings underline the potential significance of weight management, including nutritional support and dietary counseling, in enhancing outcomes for patients with *Staphylococcus aureus* sepsis. Patients in higher BMI categories often presented with higher incidences of conditions like congestive heart failure, chronic pulmonary disease, diabetes, and cerebrovascular disease and were typically younger. Numerous studies have supported the association between

being overweight or obese and improved survival, termed the "obesity paradox" [26,27]. The mechanisms behind this paradox are still elusive, with one theory suggests that the hypermetabolic state in obese individuals, accompanied by abundant glucose and fatty acids, not only promotes the production of bioactive peptides such as leptin by adipocytes but also activates immune T cells [28]. These factors may regulate the initial inflammatory response in critical illnesses by inhibiting the release of pro-inflammatory cytokines from macrophages. [29]. Another hypothesis proposes that additional energy reserves in adipose tissue might protect overweight and obese patients from the adverse effects of catabolic processes during severe illnesses, thereby improving outcomes. Moreover, these patients often exhibit higher serum cholesterol and lipid levels, which, despite long-term risks, might offer benefits during the acute phase of critical illnesses by binding endotoxins and mitigating their inflammatory effects [30].

This study effectively stratified the risk of *Staphylococcus aureus* sepsis according to BMI group, representing a strength of our approach. However, several limitations should be considered when interpreting our findings. Firstly, our results are based on post hoc analysis of publicly available databases, preventing us from establishing causality. Secondly, while the analysis adjusted for known confounders, our observations might still be influenced by residual or unmeasured factors. Additionally, we only considered initial BMI values obtained within 24 hours of ICU admission, leaving the impact of BMI fluctuations on prognosis uncertain. Lastly, given that this is a retrospective database analysis, our findings require validation through a multicenter, prospective study with a larger sample size.

## Conclusion

This cohort study demonstrated a significant association between higher BMI and reduced 28-day all-cause mortality among ICU patients with *Staphylococcus aureus* sepsis.

## Acknowledgments

We thank the MIMIC-IV database for providing the original study data.

## Author Contributions

**Conceptualization:** Heping Xu, Yiqiao Liu.

**Formal analysis:** Heping Xu, Yiqiao Liu.

**Funding acquisition:** Heping Xu.

**Investigation:** Huan Niu, Feng Zhan.

**Methodology:** Huan Niu.

**Resources:** Huan Niu, Hong Wang.

**Writing – original draft:** Heping Xu, Yiqiao Liu.

**Writing – review & editing:** Huan Niu, Hong Wang, Feng Zhan.

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
