## [Decision Letter · Decision Letter 0]

12 Jun 2024

PONE-D-24-16656Impact of body mass index on mortality outcomes in intensive care patients with staphylococcus aureus sepsis: a retrospective analysisPLOS ONE

Dear Dr. Xu,

Thank you for submitting your manuscript to PLOS ONE. After careful consideration, we feel that it has merit but does not fully meet PLOS ONE’s publication criteria as it currently stands. Therefore, we invite you to submit a revised version of the manuscript that addresses the points raised during the review process.

We look forward to receiving your revised manuscript.

Kind regards,

Eleni Magira

Academic Editor

PLOS ONE

Journal Requirements:

"This work was supported by Hainan Provincial Natural Science Foundation of China.  Project 823RC560."

"The authors declare that they have no competing interests."

5. Please ensure that you refer to Figure 2 in your text as, if accepted, production will need this reference to link the reader to the figure.

Additional Editor Comments:

Dear authors

Thank you for your submission regarding the relationship between body mass index and mortality in patients with Staphylococcus aureus sepsis using the MIMIC-IV database.

However, further clarification in the discussion is need it regarding your findings. For instance, could you try an interpretation regarding type of ventilation in obese and mortality? IS severity score like SOFA II playing a role in prognosis ?

Please see the following comments to the authors.

Reviewers' comments:

Reviewer's Responses to Questions

**Comments to the Author**

1. Is the manuscript technically sound, and do the data support the conclusions?

Reviewer #1: Partly

Reviewer #2: Partly

2. Has the statistical analysis been performed appropriately and rigorously? 

Reviewer #1: Yes

Reviewer #2: Yes

3. Have the authors made all data underlying the findings in their manuscript fully available?

Reviewer #1: Yes

Reviewer #2: Yes

4. Is the manuscript presented in an intelligible fashion and written in standard English?

Reviewer #1: Yes

Reviewer #2: Yes

5. Review Comments to the Author

Reviewer #1: In this study, the authors demonstrated the relationship between body mass index and mortality in patients with Staphylococcus aureus sepsis using the MIMIC-IV database. However, further clarification and revision of the descriptions are needed for publication as follows. Please see the following comments to the authors.

Major comments

Abstract

1. The statement in the conclusion that "non-invasive ventilation patients with a higher BMI had a lower risk of death compared to invasive ventilation patients" is not supported by the results presented in the manuscript. It is essential to ensure that the conclusions drawn are directly interpretable from the results section. Please revise or substantiate this claim with appropriate data from the results.

Introduction

1. The introduction is currently presented as a single block of text. To improve readability and ensure that the content is easily digestible, please reformat this section into several distinct paragraphs.

Methods

1. You have defined AKI based on serum creatinine levels. However, the definition is incomplete without considering changes in urine output. Please include this information to provide a comprehensive definition of AKI.

2. While you mention using the Sepsis-3 criteria for diagnosing sepsis, there is no clarification on how organ dysfunction was confirmed. Please describe the method used to extract and confirm organ dysfunction.

3. The choice of variables included in Model 3 of the multivariate regression analysis needs clarification. While including variables related to the patient's condition within 24 hours of ICU admission is understandable, the rationale behind adjusting for ICU and hospital length of stay is unclear. These variables could introduce bias related to the outcome rather than serve as confounders. Please reconsider their inclusion or provide a robust justification.

4. The inclusion of AKI should be consistent with the 24-hour post-ICU admission timeline. Additionally, the role of sex as a variable in Model 3 needs clarification regarding its relevance and impact.

5. Please explain the rationale behind setting the age threshold at 75 for subgroup analyses.

Results

1. The use of two different categorizations for BMI (numerical BMI and obesity class) is confusing and could potentially lead to misinterpretation of the data. If obesity class is to be used, please clarify its relevance to the study aims and consider moving detailed data, like those in Table 2, to supplementary content.

2. The Kaplan-Meier curve is not mentioned in the Methods section. Please ensure that all statistical methods used in the study are thoroughly described in the Methods section.

3. The mention of WHO obesity classes is more appropriate for the Results section where classification criteria directly impact the data presentation. Please relocate this information accordingly.

Minor comments

1. The first character of “staphylococcus aureus” in the title should be capitalized.

2. “Staphylococcus aureus” is usually written in italic as “Staphylococcus aureus.”

Reviewer #2: Thanks for the opportunity to review the manuscript “Impact of body mass index on mortality outcomes in intensive care patients with staphylococcus aureus sepsis: a retrospective analysis”.

The authors expand the phenomena of the “obesitas paradox” to patients with Staphylococcus aureus sepsis cared for in the ICU, more is better in terms of BMI and mortality. An impressive large cohort, over 2,000 patients, is elaborated to examine the protecting effect of rising BMI.

However, there are some remaining questions.

Major concerns

A database, MIMIC-IV, is used to extract data. However, the description of this database is scarce, for example, number and kind of ICUs? The sampling encompasses 12 years (2008-2019), what about temporal changes and impact of year of data?

Many ICD-9 and ICD-10 codes is used to incorporate patients with staphylococcus aureus. However, a validation of the codes used is not possible due to the restricted gain to the data: “All data was accessed with read-only permissions to preserve integrity.” Are only patients with a main diagnosis of Staphylococcus aureus sepsis included?

The BMI groups were alike for several variables, for as example MRSA. But does MRSA depict? MRSA bacteraemia? A local culture positive for MRSA? Please elaborate.

In the Results section it is said that the highest BMI group developed several conditions. But it is unclear if this adress condition before the admission or during the admission.

The Discussion section repeats a number of results but lacks in interpreting these. For example; is the assocation between BMI and outcome in staphylococcus aureus sepsis due to the pathogen, the organ infection usually caused by the bacterium, or is the association regardless of the pathogen, ie the same as for sepsis caused by other pathogens? Of course the findings is only assocations, but the authors should try to interpret. The Discussion section needs a reframing.

Minor concerns

The references are in general old. The authors should try to incorporate newer ones. Only one exmple to consider: Alsio¨ Å, Nasic S, Ljungstro¨m L, Jacobsson G (2021) Impact of obesity on outcome

of severe bacterial infections. PLoS ONE 16(5): e0251887. https://doi.org/10.1371/journal.

pone.0251887

Some patients do not have data for height and length. How many and is there a need for a dropout-analysis?

The association between BMI and outcome in staphylococcus aures sepsis is sometimes described as a linkage in the manuscript. Is this a proper term?

The age limit is 18 y or 19 y?

What is the rationale for including the variabel red cell distribution width (RDW)?

Why not including information about lymphocyte count and the neutrophil-lymphocyte count ratio, as it has been proposed to be a factor differentiating the inflammatory response between the obese and normalweights?

6. PLOS authors have the option to publish the peer review history of their article (what does this mean?). If published, this will include your full peer review and any attached files.

Reviewer #1: No

Reviewer #2: **Yes: **Gunnar Jacobsson

---

## [Author Response · Author response to Decision Letter 0]

8 Jul 2024

PONE-D-24-16656

Impact of body mass index on mortality outcomes in intensive care patients with staphylococcus aureus sepsis: a retrospective analysis

PLOS ONE

Dear Dr. Xu,

Thank you for submitting your manuscript to PLOS ONE. After careful consideration, we feel that it has merit but does not fully meet PLOS ONE’s publication criteria as it currently stands. Therefore, we invite you to submit a revised version of the manuscript that addresses the points raised during the review process.

We look forward to receiving your revised manuscript.

Kind regards,

Eleni Magira

Academic Editor

PLOS ONE

Journal Requirements:

"This work was supported by Hainan Provincial Natural Science Foundation of China. Project 823RC560."

"The authors declare that they have no competing interests."

5. Please ensure that you refer to Figure 2 in your text as, if accepted, production will need this reference to link the reader to the figure.

Additional Editor Comments:

Dear authors

Thank you for your submission regarding the relationship between body mass index and mortality in patients with Staphylococcus aureus sepsis using the MIMIC-IV database.

However, further clarification in the discussion is need it regarding your findings. For instance, could you try an interpretation regarding type of ventilation in obese and mortality? IS severity score like SOFA II playing a role in prognosis ?

Please see the following comments to the authors.

Reviewers' comments:

Reviewer's Responses to Questions

Comments to the Author

1. Is the manuscript technically sound, and do the data support the conclusions?

Reviewer #1: Partly

Reviewer #2: Partly

2. Has the statistical analysis been performed appropriately and rigorously?

Reviewer #1: Yes

Reviewer #2: Yes

3. Have the authors made all data underlying the findings in their manuscript fully available?

Reviewer #1: Yes

Reviewer #2: Yes

4. Is the manuscript presented in an intelligible fashion and written in standard English?

Reviewer #1: Yes

Reviewer #2: Yes

5. Review Comments to the Author

Reviewer #1: In this study, the authors demonstrated the relationship between body mass index and mortality in patients with Staphylococcus aureus sepsis using the MIMIC-IV database. However, further clarification and revision of the descriptions are needed for publication as follows. Please see the following comments to the authors.

Major comments

Abstract

1. The statement in the conclusion that "non-invasive ventilation patients with a higher BMI had a lower risk of death compared to invasive ventilation patients" is not supported by the results presented in the manuscript. It is essential to ensure that the conclusions drawn are directly interpretable from the results section. Please revise or substantiate this claim with appropriate data from the results.

Introduction

1. The introduction is currently presented as a single block of text. To improve readability and ensure that the content is easily digestible, please reformat this section into several distinct paragraphs.

Methods

1. You have defined AKI based on serum creatinine levels. However, the definition is incomplete without considering changes in urine output. Please include this information to provide a comprehensive definition of AKI.

2. While you mention using the Sepsis-3 criteria for diagnosing sepsis, there is no clarification on how organ dysfunction was confirmed. Please describe the method used to extract and confirm organ dysfunction.

3. The choice of variables included in Model 3 of the multivariate regression analysis needs clarification. While including variables related to the patient's condition within 24 hours of ICU admission is understandable, the rationale behind adjusting for ICU and hospital length of stay is unclear. These variables could introduce bias related to the outcome rather than serve as confounders. Please reconsider their inclusion or provide a robust justification.

4. The inclusion of AKI should be consistent with the 24-hour post-ICU admission timeline. Additionally, the role of sex as a variable in Model 3 needs clarification regarding its relevance and impact.

5. Please explain the rationale behind setting the age threshold at 75 for subgroup analyses.

Results

1. The use of two different categorizations for BMI (numerical BMI and obesity class) is confusing and could potentially lead to misinterpretation of the data. If obesity class is to be used, please clarify its relevance to the study aims and consider moving detailed data, like those in Table 2, to supplementary content.

2. The Kaplan-Meier curve is not mentioned in the Methods section. Please ensure that all statistical methods used in the study are thoroughly described in the Methods section.

3. The mention of WHO obesity classes is more appropriate for the Results section where classification criteria directly impact the data presentation. Please relocate this information accordingly.

Minor comments

1. The first character of “staphylococcus aureus” in the title should be capitalized.

2. “Staphylococcus aureus” is usually written in italic as “Staphylococcus aureus.”

Reviewer #2: Thanks for the opportunity to review the manuscript “Impact of body mass index on mortality outcomes in intensive care patients with staphylococcus aureus sepsis: a retrospective analysis”.

The authors expand the phenomena of the “obesitas paradox” to patients with Staphylococcus aureus sepsis cared for in the ICU, more is better in terms of BMI and mortality. An impressive large cohort, over 2,000 patients, is elaborated to examine the protecting effect of rising BMI.

However, there are some remaining questions.

Major concerns

A database, MIMIC-IV, is used to extract data. However, the description of this database is scarce, for example, number and kind of ICUs? The sampling encompasses 12 years (2008-2019), what about temporal changes and impact of year of data?

Many ICD-9 and ICD-10 codes is used to incorporate patients with staphylococcus aureus. However, a validation of the codes used is not possible due to the restricted gain to the data: “All data was accessed with read-only permissions to preserve integrity.” Are only patients with a main diagnosis of Staphylococcus aureus sepsis included?

The BMI groups were alike for several variables, for as example MRSA. But does MRSA depict? MRSA bacteraemia? A local culture positive for MRSA? Please elaborate.

In the Results section it is said that the highest BMI group developed several conditions. But it is unclear if this adress condition before the admission or during the admission.

The Discussion section repeats a number of results but lacks in interpreting these. For example; is the assocation between BMI and outcome in staphylococcus aureus sepsis due to the pathogen, the organ infection usually caused by the bacterium, or is the association regardless of the pathogen, ie the same as for sepsis caused by other pathogens? Of course the findings is only assocations, but the authors should try to interpret. The Discussion section needs a reframing.

Minor concerns

The references are in general old. The authors should try to incorporate newer ones. Only one exmple to consider: Alsio¨ Å, Nasic S, Ljungstro¨m L, Jacobsson G (2021) Impact of obesity on outcome

of severe bacterial infections. PLoS ONE 16(5): e0251887. https://doi.org/10.1371/journal.

pone.0251887

Some patients do not have data for height and length. How many and is there a need for a dropout-analysis?

The association between BMI and outcome in staphylococcus aures sepsis is sometimes described as a linkage in the manuscript. Is this a proper term?

The age limit is 18 y or 19 y?

What is the rationale for including the variabel red cell distribution width (RDW)?

Why not including information about lymphocyte count and the neutrophil-lymphocyte count ratio, as it has been proposed to be a factor differentiating the inflammatory response between the obese and normalweights?

6. PLOS authors have the option to publish the peer review history of their article (what does this mean?). If published, this will include your full peer review and any attached files.

Do you want your identity to be public for this peer review? For information about this choice, including consent withdrawal, please see our Privacy Policy.

Reviewer #1: No

Reviewer #2: Yes: Gunnar Jacobsson

---

## [Decision Letter · Decision Letter 1]

25 Jul 2024

Impact of body mass index on mortality outcomes in intensive care patients with staphylococcus aureus sepsis: a retrospective analysis

PONE-D-24-16656R1

Dear Dr. Heping Xu

We’re pleased to inform you that your manuscript has been judged scientifically suitable for publication and will be formally accepted for publication once it meets all outstanding technical requirements.

Kind regards,

Eleni Magira

Academic Editor

PLOS ONE

Additional Editor Comments (optional):

Reviewers' comments:

Reviewer's Responses to Questions

**Comments to the Author**

1. If the authors have adequately addressed your comments raised in a previous round of review and you feel that this manuscript is now acceptable for publication, you may indicate that here to bypass the “Comments to the Author” section, enter your conflict of interest statement in the “Confidential to Editor” section, and submit your "Accept" recommendation.

Reviewer #1: All comments have been addressed

2. Is the manuscript technically sound, and do the data support the conclusions?

Reviewer #1: Yes

3. Has the statistical analysis been performed appropriately and rigorously? 

Reviewer #1: Yes

4. Have the authors made all data underlying the findings in their manuscript fully available?

Reviewer #1: Yes

5. Is the manuscript presented in an intelligible fashion and written in standard English?

Reviewer #1: Yes

6. Review Comments to the Author

Reviewer #1: The authors have adequately revised the manuscript according to the reviewers' recommendations. This version of the manuscript seems suitable for publication.

7. PLOS authors have the option to publish the peer review history of their article (what does this mean?). If published, this will include your full peer review and any attached files.

Reviewer #1: No

---

## [Editor Report · Acceptance letter]

29 Jul 2024

PONE-D-24-16656R1 

PLOS ONE

Dear Dr. Xu, 

I'm pleased to inform you that your manuscript has been deemed suitable for publication in PLOS ONE. Congratulations! Your manuscript is now being handed over to our production team.

Kind regards, 

on behalf of

Dr. Eleni Magira 

Academic Editor

PLOS ONE